# Molecular Identification and Expression Analysis of an Intelectin Gene in the Yellow Catfish *Pelteobagrus fulvidraco* (Siluriformes: Bagridae)

Senhao Jiang [1,2,3] , Yuting Lei [1,2], Yanxuan Li [1], Wanyan Sun [1], Ti Wang [1], Ruiting Ma [1], Qiuning Liu [1,*] and Boping Tang [1,*]

1   Jiangsu Key Laboratory for Bioresources of Saline Soils, Jiangsu Provincial Key Laboratory of Coastal Wetland Bioresources and Environmental Protection, Jiangsu Synthetic Innovation Center for Coastal Bio-Agriculture, School of Wetlands, Yancheng Teachers University, Yancheng 224007, China; longdance@sina.com (S.J.); 18351553181@163.com (Y.L.); 18036318225@163.com (R.M.)
2   College of Fisheries and Life Science, Shanghai Ocean University, Shanghai 201306, China
3   Jiangsu Haichen Technology Group Company Limited, Nantong 226602, China
*   Correspondence: liuqn@yctu.edu.cn (Q.L.); boptang@163.com (B.T.)

**Abstract:** Intelectins (*ITLN*s) are a family of calcium-dependent lectins with carbohydrate-binding capacity, are distributed across various vertebrates, and play an important role in the innate immune response against pathogen infection. The yellow catfish *Pelteobagrus fulvidraco* (Siluriformes: Bagridae) is an economically important fish in China. The aim of this study was to quantify the gene expression of *ITLN* in response to pathogen-associated molecular patterns (PAMPs) stimulation. Here, the *ITLN* gene of *P. fulvidraco* was characterized and named *PfITLN*. The full-length cDNA of *PfITLN* was 1132 bp, including a 5′-untranslated region (UTR) of 140 bp, a 3′-UTR of 110 bp, and an open reading frame (ORF) of 882 bp encoding a polypeptide of 293 amino acids, which contains a signal peptide and two fibrinogen-related domains (FReDs). PfITLN had a molecular weight of 32.39 kDa with a theoretical pI of 5.03. The deduced PfITLN amino acid sequence had 81%, 64%, and 55% homology with *Ictalurus furcatus*, *Danio rerio*, and *Homo sapiens*, respectively. Moreover, the predicted tertiary protein structure of PfITLN was highly similar to that of other animals, and phylogenetic analysis showed that the PfITLN protein was close to those of other Teleostei. Real-time quantitative reverse transcription-PCR (qRT-PCR) analysis showed *PfITLN* expression in all examined tissues, with the highest abundance seen in the liver, followed by the head kidney, spleen, trunk kidney, and muscle. After PAMP infection with lipopolysaccharide (LPS) and polyriboinosinic polyribocytidylic acid (poly I:C), the expression levels of *PfITLN* were significantly upregulated at different time points. These results suggested that *PfITLN* might be involved in innate immunity.

**Keywords:** *Pelteobagrus fulvidraco*; intelectin; immune response; expression analysis

**Key Contribution:** An intelectin (*ITLN*) gene was cloned and detected in the liver of the yellow catfish *Pelteobagrus fulvidraco*. The expression of *P. fulvidraco ITLN* mRNA levels was quantified in a wide range of tissues and was upregulated when challenged with poly I:C or LPS.

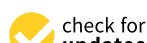



## 1. Introduction

The immune system of fish expresses both innate and acquired immunity [1]. Innate immunity is an important line of defense against bacteria, viruses, fungi, and parasites, which is mediated by pattern recognition receptors (PRRs) and can activate the adaptive immune response.  Fish have less diverse adaptive immunity compared to mammals, thereby making innate immunity more vital for fish. Scientists have only discovered several immunoglobulin isotypes and confirmed that fish do not possess a major histocompatibility complex [2].  Groups of related microorganisms have unique microbial molecules, also

known as pathogen-associated molecular patterns (PAMPs), which can be recognized by host PRRs. The primary components of microbes' cell walls are often carbohydrate chains such as LPS (lipopolysaccharides), PGN (peptidoglycan), LTA (lipoteichoic acids), and β-glucan; these carbohydrate chains are commonly found in microbial molecules [3]. LPS, a major cell wall component of the outer membrane of Gram-negative bacteria, plays a key role in host–pathogen interactions with the innate immune system. Polyinosinic-polycytidylic acid (poly I:C) functions as a synthetic counterpart of double-stranded RNA (dsRNA), initiating the activation of innate immunity against infections by triggering specific PRRs such as Toll-like receptor 3 (TLR 3) and retinoic acid-inducible gene I (RIG-I)-like receptors, which consist of RIG-I and melanoma differentiation-associated gene 5. Various PRRs include PGRPs (peptidoglycan recognition proteins), TEPs (thioester-containing proteins), GNBPs (Gram-negative bacteria-binding proteins), SCRs (scavenger receptors), lectins, GALEs (galectins), TLRs (Toll-like receptors), RIG-like receptors (retinoic acid-inducible gene-like receptors), hemoglobin, and NOD-like receptors [4]. Lectins are an important component of the innate immune system for PRRs and include one or more carbohydrate recognition domains (CRDs) that can recognize and bind to a carbohydrate on the surface of bacteria, fungi, and viruses [5]. Based on their structure, binding specificity, and calcium dependency, lectins are divided into seven different families, including C-type lectins [6], F-type lectins [7], galectins [8], intelectins (ITLNs) [9], rhamnose-binding lectins [10], I-type lectins [11], and Lily-type lectins [12].

ITLNs are a newly recognized type of glycan-binding lectin involved in many physiological and pathological processes, including polyspermy [13], immune defense induced by pathogen infections [14], immune responses induced by parasites [15], asthma [16], iron metabolism [17], cancer, and the regulation of bone density [18]. *ITLN* was first isolated and cloned in *Xenopus laevis* oocytes and named X-lectin, and discovered a closely-related gene in the small intestinal tract of mice and gave it the name "intelectin" [19]. Subsequent *ITLN* homologs have been identified in many species, including mammals [20], amphibians [21], and fish [22,23], while avian genomes lack homology. Although the sequences of ITLNs remain consistent across various species, their expression patterns, quaternary structures, and functions vary significantly both within and between species [24]. The amino acid sequences of ITLNs in vertebrates exhibit a high level of conservation, encompassing the N-terminal fibrinogen-related domain (FReD) and the C-terminal intelectin domain [25]. As vertebrates, fish act as a bridge between innate and adaptive immunity and are considered an important model in comparative immunology studies [1]. Thus far, *ITLN*s have been identified and studied in many fish species, such as blunt snout bream (*Megalobrama amblycephala*) [26], channel catfish (*Ictalurus punctatus*) [27], rainbow trout (*Oncorhynchus mykiss*) [28], common carp (*Cyprinus carpio*) [29], and grass carp (*Ctenopharyngodon idella*) [30]. The crystal structure of human *ITLN1* (*hITLN1*) and *Xenopus* embryonic epidermal lectin (XEEL) were reported, which provided novel insights into understanding their carbohydrate binding capacity and role in the innate immune response. Meanwhile, both XEEL and hITLN1 exhibit a high degree of similarity in both their natural forms and the way they bind ligands [24,25]. Several studies have confirmed that *ITLN*s play an important role in the immune response against pathogen infection and that expression levels are upregulated after pathogen challenge [31]. While the structure, expression pattern, and function of *ITLN*s are known in mammals and amphibians, information on the expression regulation and structure of *ITLN*s in *P. fulvidraco* is yet to be reported.

The yellow catfish *Pelteobagrus fulvidraco* is an important commercial freshwater aquaculture species in Asian countries, with a high market value in China. Recently, yellow catfish have been used as an experimental model for fish breeding [32], development [33], lipid metabolism [34], genomics [35], and toxicology [36]. Although infectious disease outbreaks associated with pathogenic microorganisms have caused high mortality rates and led to catastrophic economic losses in farmed *P. fulvidraco*, little is known about the immune system of this species. We previously screened a cDNA library of *P. fulvidraco* upon immune challenge [37–40] and identified an ITLN homolog that showed high similarity

with other fish ITLNs. The full-length cDNA was obtained via rapid amplification of cDNA ends (RACE)-PCR for the investigation of expression patterns in different tissues and of the immune responses. The ITLNs protein sequences from various animals were comparatively analyzed and used to align and reconstruct their phylogenetic relationship. These data provide insights into the role of ITLN in yellow catfish.

## 2. Materials and Methods

### 2.1. Sample Collection

In May 2020, yellow catfish weighing $50 \pm 10$ g were collected from Yancheng, Jiangsu province, China, and were kept at 24 °C prior to experimentation. Thirteen tissues, including blood, brain, gill, head kidney, heart, intestine, liver, muscle, ovary, skin, spleen, testis, and trunk kidney were dissected to measure the expression pattern of *PfITLN*. Thirty fish were allocated across five PVC containers and kept at 24 °C. Five fish were randomly chosen and injected with either 100 µL of phosphate-buffered saline as control or lipopolysaccharides (LPS, L-2654, Sigma, St. Louis, MO, USA) or polyribose acid (poly I:C, P9582, Sigma St. Louis, MO, USA). After treatment, head kidney, blood, liver, and spleen tissues were quickly collected after 3, 6, 12, 24, 36, and 48 h and stored at −80 °C.

### 2.2. RNA Extraction and cDNA Synthesis

Total RNA was extracted using TRIzol reagent (Sangon, Shanghai, China) based on the manufacturer's instructions. RNase-Free DNase I was used to remove contaminant genomic DNA (Promega, Madison, WI, USA). The purity and amount of extracted RNA were quantitatively measured using a NanoDrop 2000c spectrophotometer at $OD_{260}/OD_{280}$ (NanoDrop, Wilmington, DE, USA). Using the TRUEscript cDNA Synthesis Kit (Aidlab, Beijing, China) following the manufacturer's instructions, the overall RNA concentration obtained from each sample was adjusted to 1 µg, and the first-strand cDNA was generated and stored at −20 °C for later use. The SMART™ RACE cDNA amplification kit (Clontech, Terra Bella, CA, USA) was used to incorporate single-stranded cDNAs into RACE-PCR.

### 2.3. Cloning PfITLN cDNA

Expressed sequence tags (EST) encoding an ITLN homolog were isolated from *P. fulvidraco* by random sequencing and transcriptome analysis [34–37]. Oligonucleotide primers were designed using the Primer 5.0 software (www.premierbiosoft.com/primerdesign/ (accessed on 15 October 2020)) (Table 1). Primers RC3 and RC5 were used for RACE-PCR to obtain full-length cDNA with 5 min at 95 °C, followed by 5 cycles at 95 °C for 60 s, 120 s at 65 °C, and then 35 cycles at 95 °C for 30 s, 30 s at 56 °C, and 45 s at 72 °C. PCR products were examined using 1% agarose gel. After purification, PCR products were linked to the T vector (Sangon, China) and sequenced.

**Table 1.** Primers used.

| Primer No | Primer Sequences (5′-3′) | Purpose |
| --- | --- | --- |
| *RC5* | TCTTATATCTGTGCATCAGCT | RACE-PCR |
| *RC3* | TCTCTTCAAGCAATTCCCAGT | RACE-PCR |
| F1 | TGAAGGAGATGGCTCGTGGAG | qRT-PCR |
| R1 | GGGCCGTGGTTATCAGGACA | qRT-PCR |
| Actin-F | GCACAGTAAAGGCGTTGTGA | qRT-PCR |
| Actin-R | ACATCTGCTGGAAGGTGGAC | qRT-PCR |

### 2.4. Sequence Analysis of PfITLN

A BLAST search of Genbank (http://blast.ncbi.nlm.nih.gov/blast.cgi (accessed on 5 January 2021)) was conducted using the DNASTAR Lasergene 11 (Madison, WI, USA) to distinguish the open reading frame (ORF) of PfITLN and the amino acid sequence. The

Expert Protein Analysis System (ExPAsy) was used to predict the isoelectric point (pI) and molecular weight (MW) of the derived amino acid sequence using the pI/MW tool (http://web.expasy.org/compute_pi/ (accessed on 7 March 2021)). To predict the protein signal peptide, an amino acid sequence was deduced using online SignalP server tools (http://www.cbs.dtu.dk/services/SignalP/ (accessed on 7 March 2021)). The SMART software (http://smart.embl-heidelberg.de/ (accessed on 7 March 2021)) was used to predict the functional domains. The topological structure of the transmembrane protein was investigated using the TMHMM online tool (http://www.cbs.dtu.dk/services/TMHMM (accessed on 7 March 2021)). The derived PfITLN amino acid sequence was submitted to the Swiss model protein folding server (https://swissmodel.expasy.org/ (accessed on 20 March 2021)) for automatic protein structure homology modeling.

### 2.5. Homologous Alignment and Phylogenetic Analysis

ITLN amino acid sequences from diverse species were downloaded from the GenBank database (www.ncbi.nlm.nih.gov/ (accessed on 12 July 2021)) for phylogenetic analysis. These ITLN sequences included *Bubalus bubalis* (XP_006060368), *Bos taurus* (CAO77313), *Ovis aries* (ABR23345), *Homo sapiens* (AAI17226), *Pan paniscus* (XP_008962500), *Microcebus murinus* (XP_012618482), *Mus musculus* (AAI50796), *Cricetulus griseus* (XP_007618315), *Cavia porcellus* (XP_013004555), *Xenopus tropicalis* (AAH61445), *Salmo salar* (XP_014001611), *Astyanax mexicanus* (XP_007238284), *Danio rerio* (XP_001335910), *Silurus asotus* (BAL14267), *Ictalurus furcatus* (ABW07848), *Ictalurus punctatus* (ABW07846), and *P. fulvidraco*. Sequence alignments were performed utilizing the Clustal X software (https://evomics.org/resources/software/bioinformatics-software/clustal-x/ (accessed on 16 August 2021)) [41]. The amino acid and nucleotide sequence alignment were performed, and amino acid sequences of the ITLN gene were used to reconstruct the phylogenetic relationships based on the neighbor-joining method. The phylogenetic tree was constructed using the molecular evolutionary genetics analysis software MEGA 6.0 [42]. The data were analyzed using the Poisson modification, and gaps were eliminated by absolute deletion. The topological stability of adjacent trees was estimated using 1000 bootstrap replicates.

### 2.6. Quantitative Analysis of PfITLN

qRT-PCR was utilized to investigate the level of mRNA expression of the *PfITLN* gene after immune stimulation in multiple tissues and PAMPs. The expression level of the actin gene was utilized as a housekeeping control. The primers used for qRT-PCR are shown in Table 1. qRT-PCR was performed using the Mastercycler Ep Realplex thermocycler (Eppendorf, Hamburg, Germany), utilizing the SYBR Green qPCR Mix kit (Aidlab, Beijing, China). The reaction mixture (20 µL) included 10 µL of 2 × SYBR Green qPCR Mix, 1 µL of forward and reverse primers, 1 µL of cDNA, and 7 µL of RNase-free $H_2O$. The PCR protocol included 40 cycles of 50 s at 95 °C, 15 s at 55 °C, and 30 s at 72 °C. The melting point was identified as between 60 °C and 95 °C. Each individual experiment was performed three times, and the relative gene expression was measured using the methods of Livak and Schmittgen [43].

### 2.7. Data Analysis

Data are shown as mean ± standard error of the mean (SEM). One-way ANOVA tests and a $p$-value < 0.05 were used to identify significant differences.

## 3. Results and Discussion

### 3.1. Sequence Analysis of the PfITLN Gene

The *ITLN* gene was identified using RNA extracted from the liver of *P. fulvidraco*. Full-length cDNA from *PfITLN* was obtained via RT–PCR and RACE–PCR. The resulting 1132-bp cDNA sequence included a 140-bp 5′-untranslated sequence, a putative ORF of 882-bp encoding a polypeptide of 293 amino acids with two typical structural features, fibrinogen-related domains, and a 110-bp 3′ untranslated area with a 21-bp poly (A) tail.

The nucleotide and amino acid sequences of *PfITLN* are displayed in Figure 1. Based on the amino acid sequence, the molecular weight of PfITLN was predicted to be 32.39 kDa, and the equipotential point was 5.03. A protein signal peptide forecast protocol within the SMART software (Heidelberg, Germany) identified a signaling peptide, suggesting that this is a secreted protein. Motif-scan results showed that the PfITLN protein contained a N-glycosylation site, nine casein kinase II phosphorylation sites, six protein kinase C phosphorylation sites, seven N-myristoylation sites, and a tyrosine kinase phosphorylation site. As displayed in Figure 2, the forecasted tertiary structure of PfITLN protein by the Swiss model protein folding server included alpha helices, a beta-sheet, a C-terminus, and a N-terminus, indicating that ITLN has a conserved region and similar function as in human [44]. By utilizing the conventional area forecast of the SMART software, PfITLN contains two fibrinogen-related domains, which play vital roles in blood clotting, platelet aggregation, and regulation of immune activity [25,45]. This result also revealed that this protein belonged to the lectin superfamily, a diverse family of proteins involved in activity towards N-acetylglucosamine via their fibrogen-like domains that are needed in a variety of cellular processes, including blood clotting, immune response, and the regulation of neurogenesis.

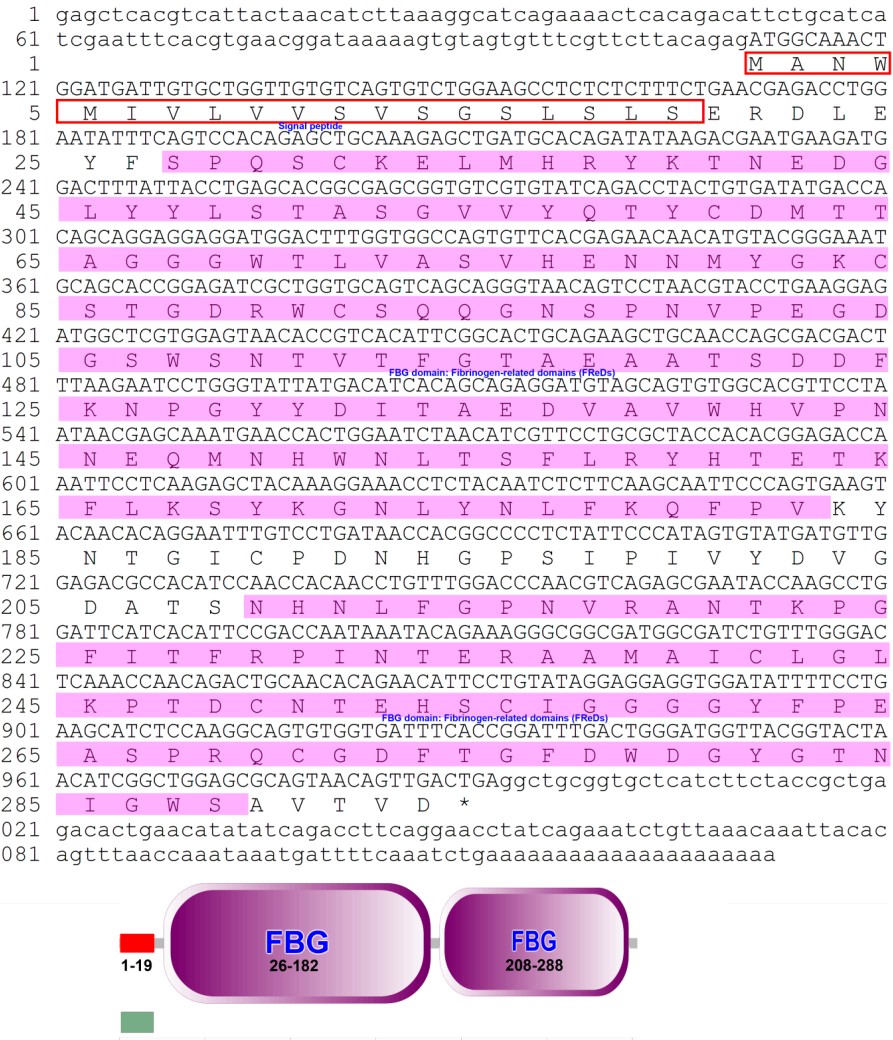

**Figure 1.** Complete nucleotide and derived amino acid sequences of the *ITLN* gene of *P. fulvidraco*. Amino acid residues are represented as letters. The open reading frame (ORF) from the start codon ATG to the stop codon TAA is capitalized. Red boxes represent the signal peptide, and pink boxes represent the fibrinogen-related domains.

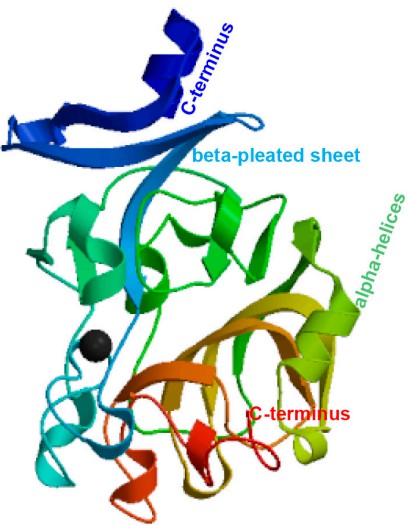

**Figure 2.** Predicted tertiary structure of the PfITLN protein.

*3.2. Homologous Sequence Alignment and Phylogenetic Analysis*

As shown in Figure 3, nine ITLN protein sequences were compared using the Clustal X software (Ballwin, MO, USA) to evaluate the evolutionary affinity of ITLN. The inferred PfITLN protein sequence was strongly consistent with the evolutionary position of the organism, with 81%, 64%, and 55% homology with *Ictalurus furcatus*, *Danio rerio*, and *Homo sapiens*, respectively. The amino acid sequence homology was over 50%, indicating that ITLN is strongly conserved during evolution [42]. Additionally, sequence alignments and functional domain predictions showed that amino acid sequences for the conserved regions of ITLN were highly similar to those in animal ITLN.

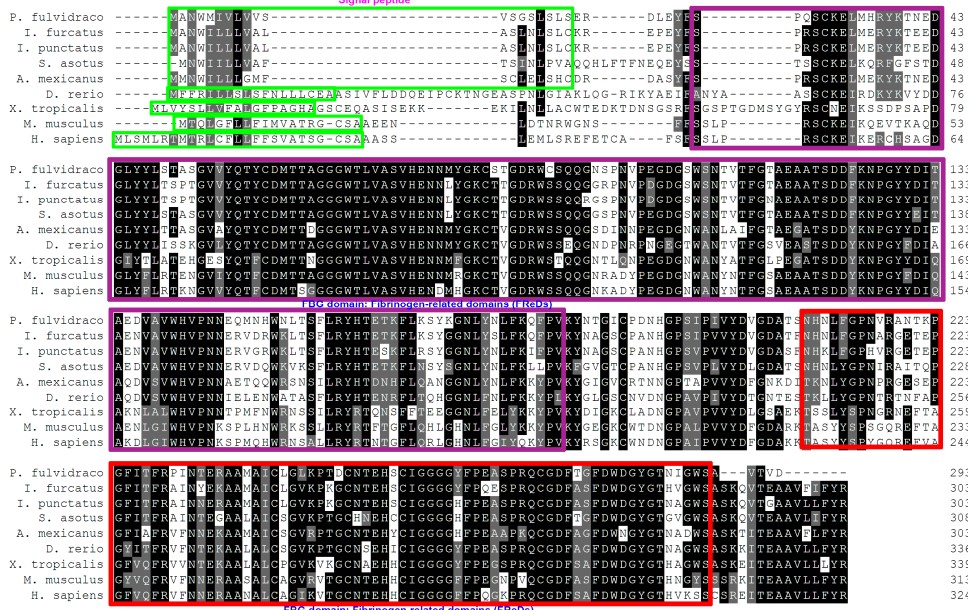

**Figure 3.** Sequence alignment of the *PfITLN* protein with its homologs. The ITLN sequences from *H. sapiens* (AAI17226), *M. musculus* (AAI50796), *Cavia porcellus* (XP_013004555), *X. tropicalis* (AAH61445), *A. mexicanus* (XP_007238284), *D. rerio* (XP_001335910), *S. asotus* (BAL14267), *I. furcatus* (ABW07848), *I. punctatus* (ABW07846), and *P. fulvidraco* are aligned by Clustal X software. Conserved amino acids are highlighted in black, and similar amino acids in grey. Purple and red boxes represent the fibrinogen-related domains, and green boxes represent the signal peptide.

We subsequently aimed to identify the correlation between configurational elements and the immune activity of ITLN. To categorize and investigate the molecular evolution of ITLN, phylogenetic relationships of 17 typical ITLN sequences were reconstructed using a neighbor-joining (NJ) approach based on the amino acid sequences. As shown in Figure 4, a total of 30 representative ITLN amino acid sequences from different organisms including *P. fulvidraco* were used to reconstruct the phylogenetic relationship. Results showed that the sequences analyzed could be grouped into four distinct categories, including mammalian, amphibian, and bony fish clades. All bony fish ITLNs were closely related within one major clade, while PfITLN was most closely related to that of *Silurus asotus*. In general, the phylogenetic tree analysis corresponded with their phylogenetic affiliations and supported the recognition of the PfITLN protein, as it has substantial homology with other phylogenetic clusters [46].

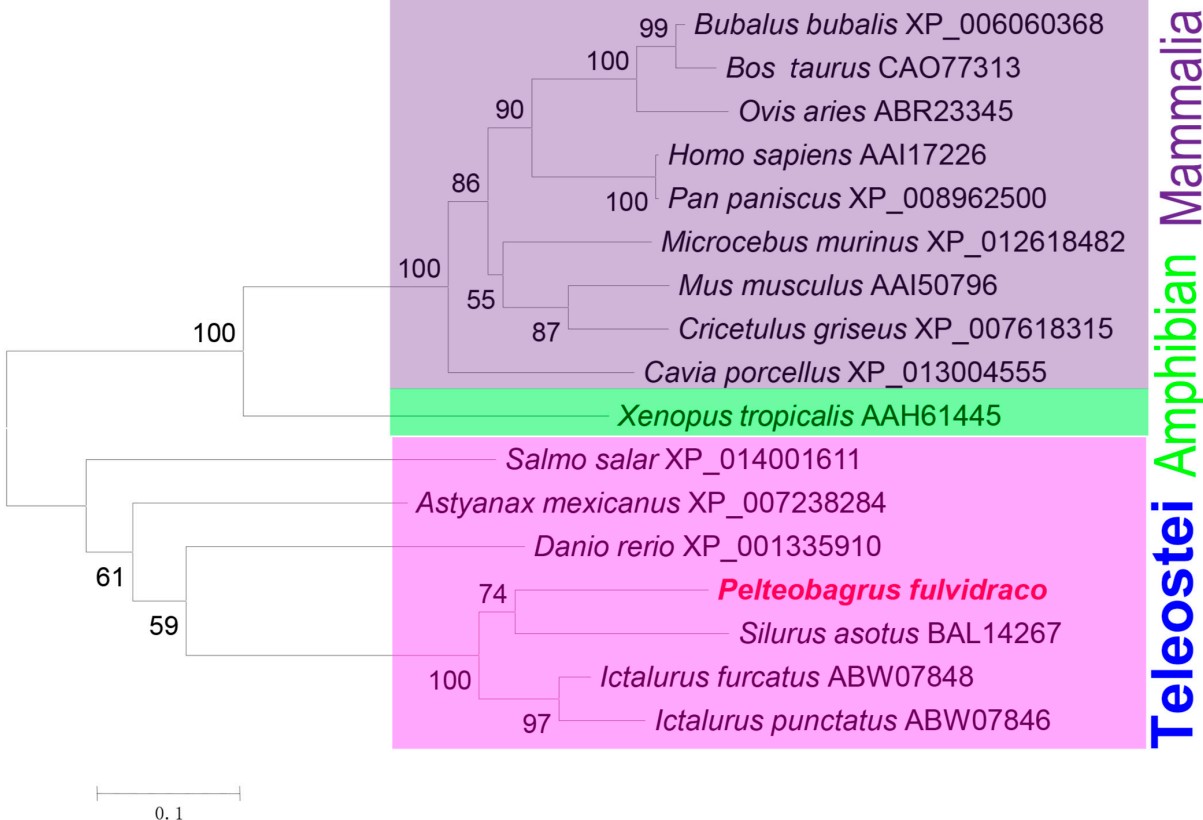

**Figure 4.** MEGA phylogenetic tree established via the NJ algorithm. Public database accession numbers of ITLN proteins are shown following the names of organisms. The bootstrap values of each branch (1000 repeats) are indicated, and the number on each node corresponds to the value.

### 3.3. Expression of the PfITLN Gene in Yellow Catfish

The relative level of expression of *PfITLN* across tissues was measured using qRT-PCR. The level of expression for each tissue was standardized to that of actin. The qRT-PCR results are shown in Figure 5 and indicate that the *PfITLN* is expressed in all investigated tissues, suggesting that the *ITLN* gene may play an important role in the development of yellow catfish. Namely, the lowest expression of *PfITLN* was detected in the blood, while the highest expression was in the liver, spleen, head kidney, trunk kidney, and muscle. The pattern of tissue expression of *ITLN* was consistent with that observed in previous studies, which also reported high expression levels in the liver, spleen, head, kidney, and intestine. Transcript levels of *ITLN* from rainbow trout were found in the sputum, liver, intestine, and skin [28]. The common carp *ITLN* gene was detected in all tissues and at high levels in the hindgut, midgut, and spleen [29]. *ITLN*s from *Ctenopharyngodon idella* were primarily

detected in the head, kidney, and spleen, as well as the intestine and gill [30]. The *ITLN* expression of *Megalobrama amblycephala* was primarily detected in the liver and spleen but also in the heart, kidney, muscle, blood, brain, intestine, and gills [47]. *ITLN2* expression was generally detected in zebrafish, with the highest level detected in the intestine and base levels in the liver [48]. In the channel and blue catfishes, the highest expression of *ITLN2* was observed in the liver, and lower levels were seen in other tissues [49]. *ITLN* was mainly expressed in the spleen, liver, and kidney of *Carassius auratus gibelio* [50]. The *ITLN* gene was found to be expressed in various tissues, including the blood, intestines, kidney, heart, gill, liver, adipose tissue, and gonads of the lamprey, *Lampetra japonica* [51]. AmphiITLN71469 showed a high level of expression in the digestive tract and skin in amphioxus [52]. AmphiITLN239631 is expressed in the muscle, epidermis, sputum, hepatic caecum, intestinal tract, and testis, with the hepatic cecum exhibiting the highest expression and the muscle displaying the lowest expression [52]. Human *ITLN1* showed a wider range of expression, including heart, small intestine, colon, kidney collecting tubule cells, bladder umbrella cells, some mesothelial cells, and follicular cells, present in both the small intestine and colon, particularly in goblet cells, with no changes in its expression observed in Crohn's disease (CD); human *ITLN2* was selectively expressed in Paneth cells of the small intestine [53,54]. Six *ITLN*s from the 129S7 mouse strain were identified with site-specific expressions in the gastrointestinal tract [55]. Taken together, *ITLN* was primarily observed in immunologically imported tissues, including the spleen, head, kidney, and liver, which suggests that *ITLN* may play an essential role in preventing microbial infection in aquatic conditions [25].

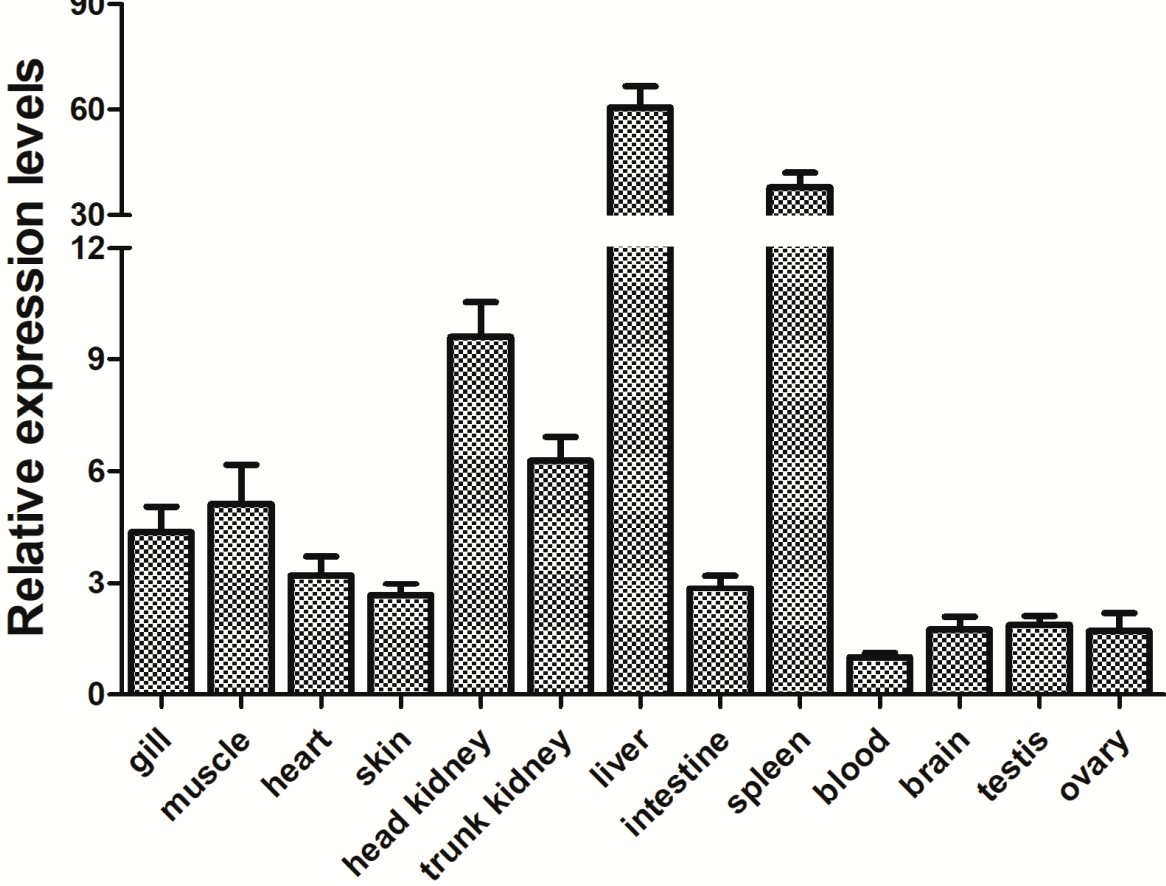

**Figure 5.** Relative expression of the *PfITLN* gene in different tissues. The actin gene was utilized as an internal control. The lowest gene expression level was set as 1.0. Data are shown as average fold-change relative to the untreated control (mean ± SE, *n* = 3).

### 3.4. Expression Profiles of PfITLN Challenged by LPS or Poly I:C

Intelectin is a lectin with the capacity to recognize and bind to carbohydrates [25]. ITLN genes play a critical role in innate immune responses to microbial infections in fish. To investigate the immune responses of *PfITLN*, the expression of *PfITLN* was investigated in immunological tissues upon antigen exposure utilizing the qRT-PCR method. Tissues were collected at the critical time points for the presentation and attachment of PAMPs (LPS and Poly I:C). As shown in Figures 6 and 7, the expression of *PfITLN* significantly upregulated in the spleen and head kidney after 3 h using LPS. Following the LPS challenge, the expression of *PfITLN* peaked at 3 h in the head kidney, at 12 h in the liver, and at 24 h in the blood, respectively. Following the poly I:C challenge, the expression of *PfITLN* was significantly upregulated after 3 h in the liver, spleen, and head kidney and peaked at 48 h in the blood. *PfITLN* expression was upregulated in other fish following LPS challenge or infection with bacteria. The spleen enables host cells to destroy invasive pathogens [56]. The expression of *ITLN* in grass carp *Ctenopharyngodon idella* was increased after LPS injection [30]. The mRNA transcript and protein levels of *ITLN* in *M. amblycephala* were dramatically upregulated in immune-related tissues at 24 h in response to the *Aeromonas hydrophila* challenge [47]. In *Danio rerio*, *ITLN2* caused bacterial agglutination and bound to the LPS or PGN of bacteria [48]. The *ITLN* expression levels of channel catfish were induced in macrophage-rich tissues injected with *Edwardsiella ictaluri* [49]. In *Branchiostoma japonicum*, *ITLN* expression was upregulated in the intestine 8 h after the *Staphylococcus aureus* challenge [57]. The *ITLN* mRNA transcript levels of rainbow trout were upregulated in response to *Listonella anguillarum* [58]. After infection with *Trichuris muris*, the small intestine of *Mus musculus* showed an increase in intelectin-2 expression [59]. The expression of *ITLN3* was significantly increased in all tissues of common carp after being infected with *S. aureus* or *A. hydrophila* [60]. Human *ITLN-1* was a host defense lectin that assisted phagocytic clearance of microorganisms and possessed the ability to bind exocyclic 1,2-diols found in the surface glycans of human pathogens such as *Streptococcus pneumoniae*, *Vibrio cholerae*, *Mycobacterium bovis* and *Helicobacter pylori* in its secreted glycoprotein form [61,62]. The co-purification of sheep intelectin-2 with mucin Muc5ac from gastric mucus suggests that intelectin might have a part in altering the consistency of mucus [63]. The expression of *ITLN3* in *X. laevis* was upregulated in the intestinal and rectal [64]. These results suggest that *ITLN* is an essential protein that plays a role in the immune response of fish.

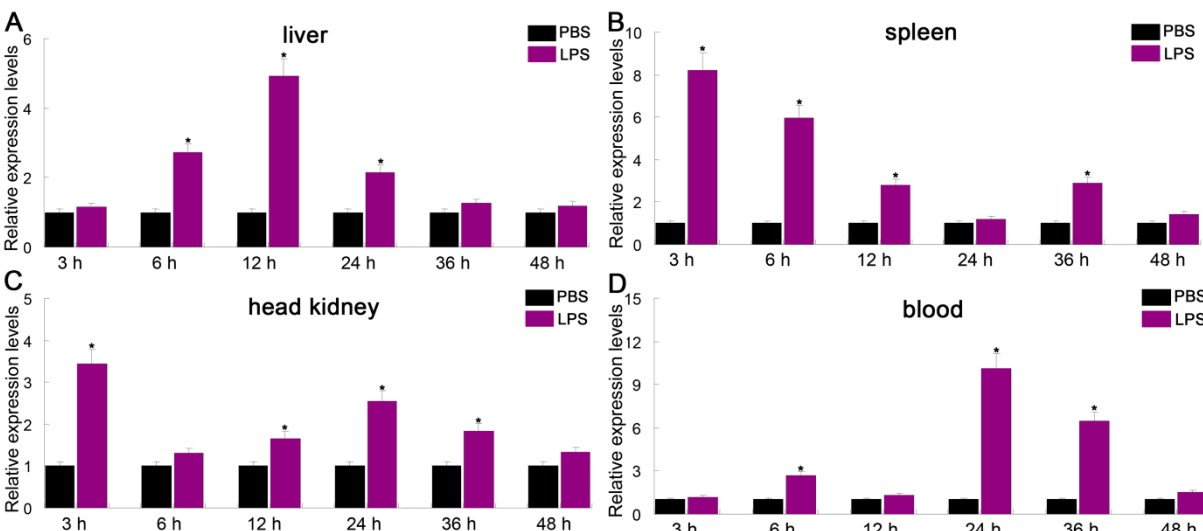

**Figure 6.** Relative expression of *PfITLN* in response to LPS in different tissues. The actin gene was utilized as an internal control. Data are shown as average fold change relative to the untreated control (mean ± SE, *n* = 3). It was considered statistically significant when *p*-values were less than 0.05 and indicated by an asterisk (*).

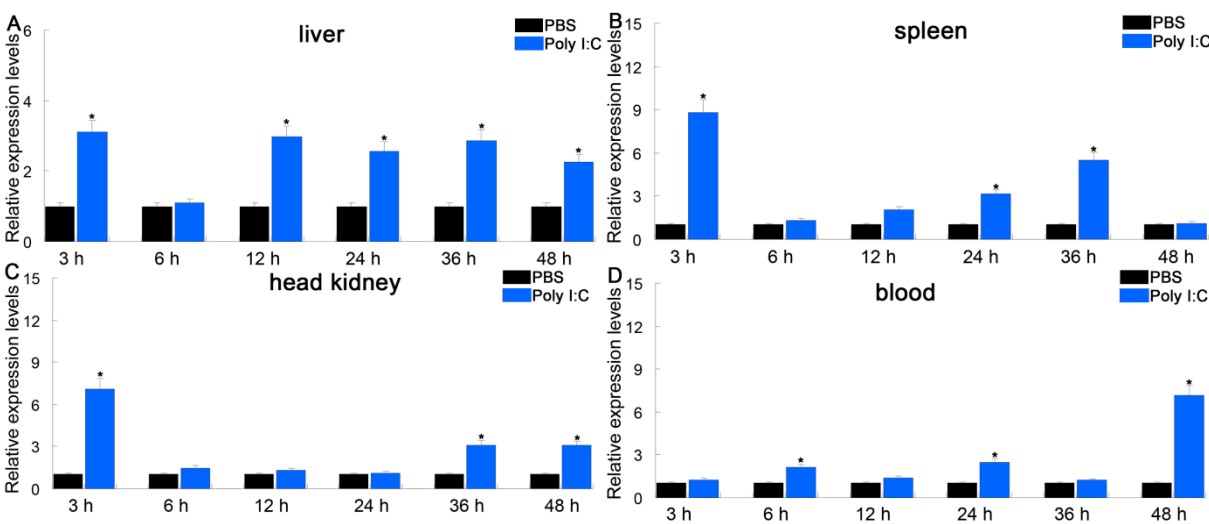

**Figure 7.** Relative expression of *PfITLN* in response to poly I:C in different tissues. The actin gene was utilized as an internal control. Data are shown as average fold-change relative to the untreated control (mean ± SE, *n* = 3). It was considered statistically significant when *p*-values were less than 0.05 and indicated by an asterisk (*).

## 4. Conclusions

Innate immunity is the primary defense mechanism in fish. ITLN, which act as pattern recognition receptors, plays a key role in the initial defense against pathogens. The ITLN gene from *P. fulvidraco* was identified and characterized, and the levels of expression were examined under the PAMPs challenge. Our results suggest the *PfITLN* may play a key role in mediating an innate immune response to PAMP exposure. Gene expression data can offer insights into the innate immune defense mechanisms and disease management and aid the development of molecular markers for disease resistance. However, further functional research should be conducted to better describe the effectiveness of using ITLNs to improve disease resistance.

**Author Contributions:** Conceptualization, Q.L. and B.T.; Data curation, Q.L.; Formal analysis, S.J., Y.L. (Yuting Lei) and R.M.; Funding acquisition, S.J., Q.L. and B.T.; Investigation, S.J., Y.L. (Yuting Lei), W.S., Q.L. and B.T.; Methodology, S.J. and Q.L.; Project administration, Q.L. and B.T.; Resources, Q.L.; Software, Q.L.; Supervision, S.J., Q.L. and B.T.; Validation, Y.L. (Yanxuan Li), W.S. and T.W.; Writing—original draft, S.J. and Q.L.; Writing—review and editing, Q.L. and B.T. All authors have read and agreed to the published version of the manuscript.

**Funding:** This work was supported by the Natural Science Research General Program of Jiangsu Provincial Higher Education Institutions (22KJA240002 and 21KJA240003); the Fishery High-Quality Development Research Project of Yancheng, China (YCSCYJ2021022); the Transverse Projects of Xiangshui Haichen Agriculture Development Company Limited (2022320907000829), the Science and Technology Vice-President Project of Jiangsu Province (FZ20230549); the National Natural Science Foundation of China (31440089, 32370556, and 32270487); the Natural Science Foundation of Jiangsu Province (BE2020673); and the National Key R&D Program of China (2019YFD0900404-05). The study was sponsored by the Qing Lan Project of Jiangsu Province and the "Outstanding Young Talents" of Yancheng Teachers University.

**Institutional Review Board Statement:** The Committee of the Yancheng Teachers University approved the animal protocols, and all experiments were performed under the applicable standards, with access No. YCTU-2020005.

**Data Availability Statement:** The data presented in this study are available upon request from the corresponding author.

**Conflicts of Interest:** The authors declare no conflict of interest.

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
