# Peer review of "Molecular Identification and Expression Analysis of an Intelectin Gene in the Yellow Catfish Pelteobagrus fulvidraco (Siluriformes: Bagridae)"

_fishes, doi:10.3390/fishes8100492_

Round 1

Reviewer 1 Report

Although it is a very common experiment, the authors should write a scientific paper in a professional manner. Several major concerns are listed as follows:

1. Language should be polished by a native speaker or professional company. The whole paper was written in a non-professional manner, including the footnote of figures. Many grammar and format errors were highlighted in the attached pdf file. Many issues in the current MS, I did not read carefully due to time limitations.

2. Lipopolysaccharide (LPS) and polyriboinosinic polyribocytidylic acid are NOT pathogens, why authors claimed the challenge as a pathogen infection?

3.  Section 2.1. (Sample collection and immune response) need to be rephrased and clarified with more details.

4. Regarding the qRT-PCR, whether the authors established standard curves? Please provide relevant data.

5. References should be kept in the same format. 

Author Response

Response to Reviewer 1 Comments

Although it is a very common experiment, the authors should write a scientific paper in a professional manner. Several major concerns are listed as follows:

  1. Language should be polished by a native speaker or professional company. The whole paper was written in a non-professional manner, including the footnote of figures. Many grammar and format errors were highlighted in the attached pdf file. Many issues in the current MS, I did not read carefully due to time limitations.

Response: Thank you for your suggestions. According to your advice, this manuscript was edited for proper English language, grammar, punctuation, spelling, and overall style by one or more of the highly qualified native English speaking editors at NativeEE. NativeEE specializes in editing and proofreading scientific manuscripts for submission to peer-reviewed journals.

  1. Lipopolysaccharide (LPS) and polyriboinosinic polyribocytidylic acid are NOT pathogens, why authors claimed the challenge as a pathogen infection?

Response: Many thanks for your valuable suggestions and comments; we made it correct in pathogen-associated molecular patterns (PAMPs).

  1. Section 2.1. (Sample collection and immune response) need to be rephrased and clarified with more details.

Response: Thanks for your very thoughtful comments, we have modified these sentence according to your suggestion.

  1. Regarding the qRT-PCR, whether the authors established standard curves? Please provide relevant data.

Response: We would like to thank the reviewer for this insightful suggestion, the standard curves was established and shown in below, green is represent the actin gene, and purple is represent the intelectin gene.

  1. References should be kept in the same format.

Response: The references were made revisions in revised manuscript according to Fishes style.

Reviewer 2 Report

     Knowledge of the innate immune system of fishes is valued for both basic and applied scientific reasons. Jiang et al. characterized the interlectin gene of P. fulvidraco. The methods are straightforward and the results seem defensible. However, the construction of the prose is so full of errors that the text is hard to follow, complicating my ability to frame a recommendation on the value of the manuscript. I have marked the manuscript to guide revision of the prose. I encourage the authors to use a professional editing service to develop a revised manuscript with much-improved prose. More context-related comments follow.

     Abstract. – At line 29, “other” must be added – the tertiary structure is much like that of OTHER animals.

     At line 29, the PfITLN GENE (or AMINO ACID?) sequence is close to THOSE OF OTHER Teleostei.

     Methods. – At lines 83-84 is an incomplete sentence. Further, I don’t see the need for the sentence here.

     At line 85, is “dry” kidney the head kidney?

     Also at line 85, surely “people” were not at issue. Were these individuals? Samples?

     At line 100, random sequencing?

     At line 101, would that be Primer 5.0 software? We need a citation or website in support of whatever software was used.

     At line 105, something is missing – a lone “t” stands at the end of the line where something more meaningful should appear.

     We need supporting citations or website addresses at lines 115, 116, 117, 122, and 130.

     At line 131, what does “ep reallx” mean? Should this be capitalized to indicate the model number of a PCR machine?

     At lines 135-139 is a passage whose intent is sufficiently unclear that I cannot understand it and offer editorial marks.  

     Results and Discussion. – The lower part of Figure 1 is not labeled. What does FBG stand for? Is the scale for counts of amino acids?

     Figure S1, if truly intended as supplemental information, should not appear in the manuscript. Actually, I feel it could be labeled as a regular figure and remain in the manuscript, but that is for the authors to decide.  

    At line 202, no, the authors did not demonstrate expression of ITLN through the life cycle, so that sentence must be recast or eliminated. They did show expression of the gene across tissues.

     At line 228, what does “for exemption” mean??

     The authors did not show expression of ILTN in response to pathogen challenge, but rather antigen challenge. The distinction is meaningful. Hence, the sentence at line 265 must be recast.  

     References. – The literature citations are close to, but not entirely in accordance with journal citation stylistics. Species Latin names should be italicized. Key words in titles of journal articles are mostly not  capitalized, and so cases where they are capitalized should be changed for consistency. I’ve marked the manuscript.

The English seems to be the result of use of an online text translator. It is not close the the standard expected. I have marked the manuscript, but with so many marks, I wonder whether the authors can follow. I encourage the authors to get the help of a professional translator.

Author Response

Response to Reviewer 2 Comments

Knowledge of the innate immune system of fishes is valued for both basic and applied scientific reasons. Jiang et al. characterized the interlectin gene of P. fulvidraco. The methods are straightforward and the results seem defensible. However, the construction of the prose is so full of errors that the text is hard to follow, complicating my ability to frame a recommendation on the value of the manuscript. I have marked the manuscript to guide revision of the prose. I encourage the authors to use a professional editing service to develop a revised manuscript with much-improved prose. More context-related comments follow.

Abstract. – At line 29, “other” must be added – the tertiary structure is much like that of OTHER animals.

Response: Thanks for review’s suggestion, “other” has been added at line 29.

At line 29, the PfITLN GENE (or AMINO ACID?) sequence is close to THOSE OF OTHER Teleostei.

Response: We have revised this sentence according to review’s suggestion.

Methods. – At lines 83-84 is an incomplete sentence. Further, I don’t see the need for the sentence here.

Response: These contents have been rewritten in the revised manuscript according to review’s suggestion.

At line 85, is “dry” kidney the head kidney?

Response: Thank you very much for pointing out the mistake. We have modified the wrong and checked in the manuscript.

Also at line 85, surely “people” were not at issue. Were these individuals? Samples?

Response: Thank you very much for pointing out the mistake. We have modified the wrong and checked in the manuscript.

At line 100, random sequencing?

Response: Thanks for review’s suggestion, we have rewritten this sentence.

At line 101, would that be Primer 5.0 software? We need a citation or website in support of whatever software was used.

Response: Thanks for review’s suggestion, primer 5.0 is a primer design application software launched by Premier, Canada. (https://doi.org/10.1089/152791600459894)

At line 105, something is missing – a lone “t” stands at the end of the line where something more meaningful should appear.

Response: Thanks! We have corrected it.

We need supporting citations or website addresses at lines 115, 116, 117, 122, and 130.

Response: Thanks for your careful review, website addresses were added in the revised manuscript.

At line 131, what does “ep reallx” mean? Should this be capitalized to indicate the model number of a PCR machine?

Response: Thank you very much for pointing out the mistake. We have corrected it.

At lines 135-139 is a passage whose intent is sufficiently unclear that I cannot understand it and offer editorial marks. 

Response: Thank you so much for the nice comment. We have modified them.

Results and Discussion. – The lower part of Figure 1 is not labeled. What does FBG stand for? Is the scale for counts of amino acids?

Response: Thanks for review’s suggestion, SMART software (http://smart.embl-heidelberg.de/) predicted two fibrinogen (FBG)-related domains at position 26-182 and 208-288 of amino acids.

Figure S1, if truly intended as supplemental information, should not appear in the manuscript. Actually, I feel it could be labeled as a regular figure and remain in the manuscript, but that is for the authors to decide. 

Response: Thanks for review’s suggestion, Figure S1 was added in the revised manuscript.

At line 202, no, the authors did not demonstrate expression of ITLN through the life cycle, so that sentence must be recast or eliminated. They did show expression of the gene across tissues.

Response: Thanks for review’s suggestion, we have rewritten this sentence.

At line 228, what does “for exemption” mean??

Response: Thank you very much for pointing out the mistake. We have corrected it.

The authors did not show expression of ILTN in response to pathogen challenge, but rather antigen challenge. The distinction is meaningful. Hence, the sentence at line 265 must be recast.

Response: These contents have been rewritten in the revised manuscript according to review’s suggestion.

References. – The literature citations are close to, but not entirely in accordance with journal citation stylistics. Species Latin names should be italicized. Key words in titles of journal articles are mostly not  capitalized, and so cases where they are capitalized should be changed for consistency. I’ve marked the manuscript.

Response: The references were made revisions in revised manuscript according to Fishes style.

The English seems to be the result of use of an online text translator. It is not close the the standard expected. I have marked the manuscript, but with so many marks, I wonder whether the authors can follow. I encourage the authors to get the help of a professional translator.

Response: Thanks so much for reviewer’ work. According to your advice, this manuscript was edited for proper English language, grammar, punctuation, spelling, and overall style by one or more of the highly qualified native English speaking editors at NativeEE. NativeEE specializes in editing and proofreading scientific manuscripts for submission to peer-reviewed journals.

Round 2

Reviewer 1 Report

No more concerns.

Author Response

Thanks for review’s suggestion, the comments are all valuable and very helpful for revising and improving our paper.

Reviewer 2 Report

     Knowledge of the innate immune system of fishes is valued for both basic and applied scientific reasons. Jiang et al. characterized the interlectin gene of P. fulvidraco. The methods are straightforward and the results defensible. This is a revised manuscript and much more readable than before. Further correction and polishing of the prose is yet needed, but I feel that the manuscript will soon prove acceptable for publication in the journal. I have marked the manuscript to guide revision of the prose. Context-related comments follow.

     Abstract. – At line 21, the “infection” was simulated using antigens, so use of “infection” is inappropriate. “Stimulation” is the appropriate word to use here.  

     Methods. –  At line 95, something is missing. Was the RNA concentration ADJUSTED TO 1 ug? And it was first-STRAND cDNA, right?

     At line 103, it must be made clear in the description of the PCR protocol how many times each amplification step was repeated.

     At line 132, not “infection”, but rather “stimulation”,  

     Results and Discussion. – At line 159, I think that the authors mean to write not of “third framework”, but rather of “tertiary structure”.

     At line 161, I think the authors need to add “as in human” for the sentence to be defensible.

     At line 172, the authors now need to refer to Figure 4, and there are now nine ITLN protein sequences.

     At lines 177 and 182, I think we have a poor translation. It is not “development” that is at issue, but rather “evolution”.

     At line 188, the authors write of Silurus lotus, but in the figure, it is shown as Silurus asotus; which is correct?

     At line 188, the authors write in error that the PfITLN gene was most evolutionarily related to that of Silurus sp., followed by D. rerio and H. sapiens. This is simply incorrect. The P. fulvidraco gene at issue was more closely related those of other fishes, then to a whole clade with an amphibian and other mammals including H. sapiens.

     At line 228, the authors should not write of “pathogen spread” – this was a simulated infection – but rather of “antigen exposure”. Similarly, at line 229, we are not speaking of “entry”, but rather of “presentation”.

     References. – The literature citations are much cleaner than before. I’ve marked the few instances where corrections are needed.

Much improved, though further work is needed.

Author Response

Knowledge of the innate immune system of fishes is valued for both basic and applied scientific reasons. Jiang et al. characterized the interlectin gene of P. fulvidraco. The methods are straightforward and the results defensible. This is a revised manuscript and much more readable than before. Further correction and polishing of the prose is yet needed, but I feel that the manuscript will soon prove acceptable for publication in the journal. I have marked the manuscript to guide revision of the prose. Context-related comments follow.

     Abstract. – At line 21, the “infection” was simulated using antigens, so use of “infection” is inappropriate. “Stimulation” is the appropriate word to use here. 

Response: Thank you very much for pointing out the mistake. We have corrected it.

     Methods. –  At line 95, something is missing. Was the RNA concentration ADJUSTED TO 1 ug? And it was first-STRAND cDNA, right?

Response: Thanks for review’s suggestion, we have revised this sentence according to review’s suggestion.

     At line 103, it must be made clear in the description of the PCR protocol how many times each amplification step was repeated.

Response: These contents have been rewritten in the revised manuscript according to review’s suggestion.

     At line 132, not “infection”, but rather “stimulation”, 

Response: Thanks for review’s suggestion, we have modified the wrong and checked in the manuscript.

     Results and Discussion. – At line 159, I think that the authors mean to write not of “third framework”, but rather of “tertiary structure”.

Response: We have revised this sentence according to review’s suggestion.

At line 161, I think the authors need to add “as in human” for the sentence to be defensible.

Response: Thanks for review’s suggestion, we have added “as in human” in the sentence.

     At line 172, the authors now need to refer to Figure 4, and there are now nine ITLN protein sequences.

Response: Thank you very much for pointing out the mistake. We have modified the wrong in the manuscript.

     At lines 177 and 182, I think we have a poor translation. It is not “development” that is at issue, but rather “evolution”.

Response: Thank you very much for pointing out the mistake. We have modified the wrong in the manuscript.

     At line 188, the authors write of Silurus lotus, but in the figure, it is shown as Silurus asotus; which is correct?

Response: Thank you very much for pointing out the mistake. We have modified the wrong in the manuscript.

     At line 188, the authors write in error that the PfITLN gene was most evolutionarily related to that of Silurus sp., followed by D. rerio and H. sapiens. This is simply incorrect. The P. fulvidraco gene at issue was more closely related those of other fishes, then to a whole clade with an amphibian and other mammals including H. sapiens.

Response: These contents have been rewritten in the revised manuscript according to review’s suggestion.

     At line 228, the authors should not write of “pathogen spread” – this was a simulated infection – but rather of “antigen exposure”. Similarly, at line 229, we are not speaking of “entry”, but rather of “presentation”.

Response: Thank you very much for pointing out the mistake. We have corrected it.

     References. – The literature citations are much cleaner than before. I’ve marked the few instances where corrections are needed.

Response: Thanks for review’s suggestion, the references were made revisions in revised manuscript according to Fishes style.